# Development of a Scale for COVID-19 Stigma and Its Psychometric Properties: A Study among Pregnant Japanese Women

**DOI:** 10.3390/bs12080257

**Published:** 2022-07-27

**Authors:** Toshinori Kitamura, Asami Matsunaga, Ayako Hada, Yukiko Ohashi, Satoru Takeda

**Affiliations:** 1Kitamura Institute of Mental Health Tokyo, Tokyo 151-0063, Japan; asamim.pn@tmd.ac.jp (A.M.); hada@institute-of-mental-health.jp (A.H.); y-ohashi@jiu.ac.jp (Y.O.); 2Kitamura KOKORO Clinic Mental Health, Tokyo 151-0063, Japan; 3T. and F. Kitamura Foundation for Studies and Skill Advancement in Mental Health, Tokyo 151-0063, Japan; 4Department of Psychiatry, Graduate School of Medicine, Nagoya University, Nagoya 464-0814, Japan; 5Department of Mental Health and Psychiatric Nursing, Tokyo Medical and Dental University, Tokyo 113-0034, Japan; 6Department of Community Mental Health and Law, National Institute of Mental Health, National Center of Neurology and Psychiatry, Tokyo 187-0031, Japan; 7Department of Nursing, Faculty of Nursing, Josai International University, Togane 283-0002, Japan; 8Department of Obstetrics & Gynecology, Faculty of Medicine, Juntendo University, Tokyo 113-8421, Japan; stakeda@juntendo.ac.jp; 9Aiiku Research Institute for Maternal, Child Health and Welfare, Imperial Gift Foundation Boshi-Aiiku-Kai, Tokyo 106-0047, Japan

**Keywords:** COVID-19, stigma, factor structure, measurement and structural invariance, construct validity

## Abstract

Background: Stigma towards COVID-19 may negatively impact people who suffer from it and those supporting and treating them. Objective: To develop and validate a scale to assess 11-item COVID-19–related stigma. Methods: A total of 696 pregnant women at a gestational age of 12 to 15 weeks were surveyed using an online survey with a newly developed scale for COVID-19 stigma and other variables. The internal consistency of the scale was calculated using omega indices. We also examined the measurement invariance of the scale. Results: Exploratory factor analyses (EFAs) of the scale items were conducted using a halved sample (*n* = 350). Confirmatory factor analyses (CFAs) among the other halved sample (*n* = 346) compared the single-, two-, three-, and four-factor structure models derived from the EFAs. The best model included the following three-factor structure (χ2/df = 2.718, CFI = 0.960, RMSEA = 0.071): Omnidirectional Avoidance, Attributional Avoidance, and Hostility. Its internal consistency was excellent (all omega indices > 0.70). The three-factor structure model showed configuration, measurement, and structural invariances between primiparas and multiparas, and between younger (less than 32 years) and older women (32 years or older). Fear of childbirth, mother–fetal bonding, obsessive compulsive symptoms, depression, adult attachment self-model, and borderline personality traits were not significantly correlated with the Omnidirectional Avoidance subscale but correlated with the Attributional Avoidance and Hostility subscales (*p* < 0.001). Conclusion: The findings suggested that our scale for COVID-19 stigma was robust in its factor structure, as well as in construct validity.

## 1. Introduction

Infectious diseases sometimes induce stigma in the general public. Historically, patients have experienced stigma due to infection with disease. Currently, researchers and practitioners share concerns regarding the stigma towards COVID-19 [1,2]. Stigma towards an infectious disease imposes an additional burden on patients who suffer from it. Furthermore, people who are stigmatised often develop self-stigma, which is the internalisation of external stigma that leads to declining self-efficacy and self-esteem for the stigmatised [3,4]. Bagcchi [5] reported that people suffering from COVID-19 are subjected to stigma, such as abandonment by their family members or the public whereas healthcare workers experience social ostracism and even attacks. This is a phenomenon observed worldwide [6,7,8]. Such stigma disrupts effective interventions and can even lead to a loss of the control of the pandemic. Stigmatising attitudes may also cause psychological distress among people suffering from COVID-19 as well as those caring for and supporting them. While the evolutionary psychological perspective indicates that infectious disease stigmatisation is adaptive for the survival and protection of the community, stigma no longer serves such a function in modern societies [9].

The COVID-19 pandemic in Japan has made people fearful of infection. In a close-knit society such as Japan, people are increasingly sensitive to community behaviour. An individual who does not do as others do may be “left out” or even seen as an “outlaw”. Such a person may be advised or even criticised in public. People without legal responsibility have been reported to provide harsh advice to citizens who they think do not conform to the social norm. Such demands of the societal norm may come from stigma in society.

Pregnant women are likely to experience psychological difficulties during the COVID-19 pandemic. Research has shown that they often suffer from mood, anxiety, and trauma symptoms (e.g., [10,11]). Stigma that pregnant women have towards the infection and infected people may negatively impact their psychological adjustment.

To control COVID-19 effectively and minimise adverse psychological effects, an examination of the influence of stigma is of primary importance. To address this issue, the development of an easy-to-use and psychometrically standardised measure of stigmatisation towards COVID-19 is a necessary first step [12]. When searching PubMed with ((stigma) AND (COVID-19)) AND (Scale OR inventory OR Measurement)), a total of 339 papers were returned. Of these, we identified 14 papers that delt with COVID-19 stigma. However, none of them treated the issue in psychometric detail (e.g., content validity, factor structure, and measurement invariance).

We reported the development and validation of a self-report scale for assessing the degree of the stigma. The data derived from a research project investigating the influences of COVID-19 on health behaviour and mental health in pregnant women in Japan. Understanding the degree of stigma towards COVID-19 by the general public using this scale would contribute to efforts to decrease the stigma and its adverse effects.

## 2. Methods

### 2.1. Study Procedures and Participants

The participants of this internet study were 696 pregnant women at 12 to 15 weeks’ gestational age. Participants were recruited for two weeks, from 7–21 December 2020, via internet application by LunaLuna and Luna Luna Baby (MTI Ltd., Tokyo, Japan). The participants were from across almost all prefectures in Japan. Anonymity was assured, and participation was voluntary. The questionnaire contained an information page, with the aims of the study, affiliations, information about informed consent, and the address of the consultation desk for the research provided. As an incentive, participants received electronic money which could be used for Amazon shopping. In order to examine structural, measurement, and structural invariances of the factor structure of the scale, we sent an e-mail to invite 696 pregnant women to participate in a follow-up study about 10 weeks later. Of the pregnant women, 245 (35.2%) responded to it.

### 2.2. Measurements

COVID-19 stigma: We developed a scale based on theoretical considerations. This instrument consists of 11 items (Table 2). Each item was rated with a 7-point Likert scale from not at all true = 0 to very much true = 6. The original questionnaire was in Japanese. Items were translated into English (see Table 2) and this was retranslated into Japanese by an individual who was unaware of the original wording, to verify the content.

Preventive means against COVID-19: We asked about the use of 7 means of infection prevention (mask, hand washing, gargling, showering, alcohol disinfection, gloves, and face guard) with a 7-point scale consisting of not at all = 0; once or twice a month = 1; once a week = 2; a few times a week = 3; once a day = 4; a few times a day = 5; and several times a day = 6.

Demographic and obstetric variables: We asked (a) the participant’s age, (b) gestational age in weeks, (c) number of past pregnancies, (d) number of past deliveries, (e) educational level, (f) infertility treatment and its types and duration in years, (g) occupational status (full- or part-time or no job), and (h) marital status.

Attitude towards the present pregnancy: With regard to the attitude towards the current pregnancy, we asked how happy or unhappy the participant was when she became aware of the pregnancy (denial of pregnancy) and whether the pregnancy was desired (intended pregnancy), both with a 5-point scale. A higher score indicated a greater denial of pregnancy and unintended (unwanted) pregnancy, respectively. In addition, we asked if they were unwilling to care for the baby after childbirth (quit caring) and if they wished to terminate the current pregnancy (wish to terminate) both with a 7-point scale.

Current pregnancy: Regarding the current pregnancy, we measured the severity of emesis using the Japanese version [13] of the 24 h Pregnancy-unique Quantification of Emesis (PUQE-24; [14,15]). This consists of only three items ((a) nausea (the length of nausea in hours for the last 24 h), (b) vomiting (number of vomiting episodes in the last 24 h), and (c) retching (the number of retching episodes in the last 24 h)) with a 5-point scale. Higher scores indicate more severe nausea and vomiting during pregnancy. 

In addition, we asked how much the current pregnancy influenced the participant (perceived impact of pregnancy). The scores were between +100 and −100 with positive scores indicating that the pregnancy was good, joyful, and happy and negative scores indicating that the pregnancy was awful, perplexing, and unhappy.

We asked whether the participant had undergone infertility treatment and, if so, about the duration of the treatment in years.

Fear of childbirth: We used the Japanese version [16] of the Wijma Delivery Expectancy/Experience Questionnaire (WDEQ; [17]). This consists of 33 items with a 5-point scale. Higher scores indicate more severe fear of the forthcoming delivery. In this study, item 31 was erroneously deleted.

Foetal bonding: We used the short version [18] of the Scale for Parent-to-Baby Emotion (SPBE; [19]). This consists of 20 items with a 7-point scale. It has 10 subscales including six basic emotions (Happiness, Anger, Fear, Sadness, Disgust, and Surprise) and four self-conscious emotions (Shame, Guilt, Alpha Pride, and Beta Pride). Each item was preceded by the question “How much do you feel the following emotion when you think of your baby in the womb?” 

Substance use: The following two ad hoc items were used to assess the frequency of tobacco and alcohol use: “How many cigarettes did you smoke before pregnancy?” and “Did you drink alcohol before pregnancy?” (Yes/No). 

Obsessive compulsive symptoms: We used the Japanese version [20] of the Obsessive Compulsive Inventory-Revised (OCI-R; [21]). This consists of 18 items with a 7-point scale. It has the following six subscales: Washing, Checking, Ordering, Obsessing, Hoarding, and Neutralising.

Depression: We used two items asking about the first two items of Major Depressive Episode (MDE), namely depressed mood and lack of interest. Each item was rated with a 4-point scale: none = 0, a few days a week = 1, more than half a week = 2, and almost every day = 3. Research showed that a set of the two questions could predict MDE reasonably well [22,23,24,25,26,27,28,29].

Adult attachment: We used the Japanese version [30] of the Relationship Questionnaire (RQ; [31]). The RQ consists of four items and a 7-point scale (does not apply to me at all = 0 to applies to me very much = 6). They indicate the following different styles of adult attachment: Secure, Fearful, Preoccupied, and Dismissing. We created the following two subscales: Positive Self- and Positive Other-models according to Bartholomew and Horowitz [31]. These were calculated as follows:Positive Self-model = Secure − Fearful − Preoccupied + Dismissing
Positive Other-model = Secure − Fearful + Preoccupied − Dismissing

Borderline personality traits: We used the short version [32] of the Personality Organisation Inventory (IPO; [33]). This consists of nine items with a 7-point scale. It has the following three subscales: Primitive Defence (PD), Identity Diffusion (ID), and Reality Testing (RT) Disturbance. 

### 2.3. Data Analysis

The whole sample was divided into the following two groups randomly: one for EFA (*n* = 350) and another for CFA (*n* = 346). Using the data from the first group, we calculated mean, SD, skewness, and kurtosis of each scale item. The Kaiser–Meyer–Olkin (KMO) index and Bartlett’s sphericity were tested as a means of assessing the factorability of the data [34]. Then, a series of exploratory factor analyses (EFAs) were performed. The maximum-likelihood method with PROMAX rotation was adopted. The model comparison of a single-, two-, three-, and four-factor structure was examined. To compare the EFA-derived factor models, we used the data from the second halved sample and performed a series of confirmatory factor analyses (CFAs) as cross-validation [35,36,37]. Starting with the single-factor model, the next model was subsequently judged as accepted if *χ*^2^ decreased significantly for the difference of *df*. This was examined repeatedly until we reached the best model. The absolute fit of the models was evaluated in terms of chi-squared, comparative fit index (CFI), and root mean square of error approximation (RMSEA). A good fit is suggested by *χ*^2^/*df* < 2, CFI > 0.97, and RMSEA < 0.05, and an acceptable fit by *χ*^2^/*df* < 3, CFI > 0.95, and RMSEA < 0.08 [38,39]. We also examined Akaike Information Criterion (AIC; [40]) of which *lower* scores indicate a better model.

The internal consistency of the model was calculated using ω. The omega coefficient is a preferable index of internal consistency of a psychological measure when the scale consists of more than one factor [41,42,43]. The proportion of variance of all the items explained by all the factors was computed as follows:$$\omega =\frac{{\left(\sum \lambda_{group1}\right)}^{2}+{\left(\sum \lambda_{group2}\right)}^{2}+{\left(\sum \lambda_{group3}\right)}^{2}}{{\left(\sum \lambda_{group1}\right)}^{2}+{\left(\sum \lambda_{group2}\right)}^{2}+{\left(\sum \lambda_{group3}\right)}^{2}+\sum_{1}^{k}\delta }\;$$
where there are three group factors. *λ* and *δ* refer to the factor loading and the unique variance of the item, respectively. The proportion of variance of the items belonging to each group factor explained by parameter estimates for the specific group factor is calculated as follows:$$\omega_{group1}=\frac{{\left(\sum \lambda_{group1}\right)}^{2}}{{\left(\sum \lambda_{group1}\right)}^{2}+\sum \delta }\;$$

After model comparison, the best model’s measurement and structural invariances were examined across different attributes (parity and age) using data from the whole sample. Starting from configuration invariances, through to metric, scalar, residual, factor variance, and factor covariance invariances to factor mean invariance were examined. The progress from one step to the next was judged as accepted if (a) the *χ*^2^ decrease was not significant for the *df* difference, (b) the decrease of CFI was less than 0.01, or (c) the increase in RMSEA was less than 0.015 [44,45]. This procedure was applied because an *χ*^2^ decrease is strongly sensitive to sample size (*N*) and, particularly in the case of a large sample, produces an unreasonable rejection of invariance.

The subscale scores were calculated by adding the scores of items derived from the factor analyses. The subscale scores were correlated with the scores of the other variables described in the measurement. The alpha level was set at *p* < 0.001 because of multiple comparisons.

### 2.4. Ethical Considerations

This study was approved by the Institutional Review Board (IRB) of the Kitamura Institute of Mental Health Tokyo (No. 2020101501). All participants provided their electronic informed consent after understanding the study rationale and procedure. The authors assert that all procedures contributing to this study comply with the ethical standards of the National and Institutional Committees on human experimentation and with the Helsinki Declaration of 1975 as revised in 2008.

## 3. Results

### 3.1. Characteristics of the Participants

The mean (SD) age of the participants was 31.7 (4.5) years, and the mean (SD) gestational age was 13.4 (1.14) weeks (Table 1). For about half of the women, the current pregnancy was their first experience. About three-quarters of the women (73.6%) were nulliparae and 26.4% were multiparae. About 90% of the women had a job at the time of the investigation. Most of them had a partner (99%). One-third of them had received infertility treatment. 

### 3.2. Scale Items

Mean, SD, skewness, and the kurtosis of 11 items of the scale in the first halved sample are shown in Table 2. None of the items showed a skewness of greater than 2. Kurtosis was less than 4 in all the items. 

### 3.3. EFA

The factorability of this data was examined with KMO = 0.837 and Bartlett’s sphericity test *χ*^2^ (55) = 1864.463, *p* < 0.001. Therefore, we performed EFAs (Table 3). In the single-factor model, all the items except items 1, 2, and 3 showed factor loading > 0.30 [46]. In the two-factor model, items 2, 6, 7, 8, 9, and 11 loaded on the first factor, and all other items except the first factor; items 6, 7, and 8 loaded on the second factor; and items 1, 2, 9, and 11 loaded on the third factor. In the four-factor model, however, only two items loaded on the second, third, and fourth factor with 0.3 or more. Thus, the four-factor model was found to be unstable as a measurement model.

### 3.4. CFA

For the cross-validation of the EFA-derived factor models, we performed CFAs with maximum likelihood mean adjusted (MLM) using the other halved sample. Single-, two-, and three-factor models were compared in terms of the goodness-of-fit (Table 4). The goodness-of-fit of the model was significantly and increasingly better from the single- to three-factor models. Thus, the three-factor model showed even better fit than any other models: *χ*^2^/*df* = 2.718, CFI = 0.960, RMSEA = 0.0.071 (Figure 1). The absolute values of goodness-of-fit were acceptable at CFI > 0.95 and RMSEA< 0.8. The first factor was loaded by items representing the avoidance of COVID-19 infected people and their family members. Nevertheless, because a substantial portion of infection symptoms were subclinical (asymptomatic), people were unaware of those who are COVID-19 positive. Thus, their avoidance and possible fear of infection were not targeted against a specific group of individuals but were more general and widespread. We termed this Omnidirectional Avoidance. The second factor was loaded by items representing avoidance of specific people who are, allegedly, more prone to infection such as medical and service workers and those who attend a clinic. Their avoidance and possibly fear towards infection were characterised by demographic features of the target of stigma (e.g., occupation). We termed this Attributional Avoidance. The third factor was loaded by items representing reproaching infected people. This may be a reflection of the participants’ hostile and potentially accusatory attitude towards those who are ‘carelessly’ infected by SaRS-COV2. We named this Hostility. We then calculated MacDonald’s *ω*. It was 0.71 for the whole scale and 0.70, 0.84, and 0.91 for Omnidirectional Avoidance, Attributional Avoidance, and Hostility, respectively. Therefore, the internal consistency was excellent.

### 3.5. Measurement Invariance

The comparison between nulliparae and multiparae (Table 5), and between the younger (less than 32) and older (32 or more) age groups (Table 6) showed that the three-factor model was invariant from the configuration, metric, scalar, factor variance, and factor covariance perspective. The factor mean for the three-factor model did not show significant differences (Table 7). This factor model was also invariant in terms of configural, measurement, and structural aspects between the two observation times (the first and second trimesters) (Table not shown).

### 3.6. Construct Validity

The scores of the three subscales―Omnidirectional Avoidance, Attributional Avoidance, and Hostility―were correlated differently from the other variables (Table 8). Omnidirectional Avoidance was significantly correlated only with the Washing scores of the OCI-R. On the other hand, the Attributional Avoidance scores were linked to fear of childbirth, all of which are subscales of the OCI-R, MDE, poor self-model, and the total score of the IPO-SV. The correlations of Hostility with the other variables were similar to those of Attributional Avoidance; however, they were also associated with negative emotions towards the foetus.

## 4. Discussion

To the best of our knowledge, our study was the first to develop a COVID-19–specific stigma scale. It consists of three independent subscales, and the three-factor structure was stable in terms of configuration, measurement, and structural invariances. Factor means did not differ in terms of parity and age. The three subscales derived from factor analyses were differently related with the other variables.

Stigma towards an illness and those people suffering from it is an attitude that appears in different domains. It may be expressed in people’s avoidant behaviours of the target illness. The target illness may, however, be difficult to recognise in such cases where the illness or suffering of people is not easily identifiable. COVID-19 is one such case. There are many cases of subclinical infection that show no observable signs or symptoms. People fear infection but find it difficult to determine whom or where they should avoid. This results in general fear of getting close to an unidentifiable target. This is represented by Omnidirectional Avoidance. Second, people learn from media, regardless of its truthfulness, that there are some groups of people who are more likely to be virus positive. These include people working at bars and restaurants, hospitals, clinics, and those attending a medical institution. The target to be avoided is clear in such cases. This is represented by Attributional Avoidance. Third, the fear of infection presents as aggression towards people who are suffering. Some people resent having recovered people return to their workplace. They no longer want to communicate with recovered people. They may claim that people who become infected are careless and therefore responsible for the spread of COVID-19. They may even claim that they should apologise. This is represented by Hostility. Although these three factors are correlated with each other to some extent, they are nevertheless independent.

We speculate that the three domains of COVID-19 stigma have different causes and consequences. Pregnant women indicating Attributional Avoidance and Hostility were more likely to show all aspects of obsessive compulsive symptoms, MDE, and borderline personality traits. Such psychopathology may lead to prejudice or vice versa. Longitudinal studies may clarify causality. These women’s marital relationship was characterised by a poor self-model. Feeling that they are not worthy of being loved may increase fear and anger towards infected people. Women high in Hostility are more likely to show borderline personality traits. We speculate that such personality traits underlie stigma and prejudice. Pregnant women expressing high Attributional Avoidance and Hostility were more likely to be fearful about the forthcoming childbirth. Women expressing high Hostility were more likely to express negative emotions towards their foetus. Tokophobia (e.g., [16,47,48]) and foetal emotional bonding (e.g., [49,50]) are very important health issues in perinatal care. Clinicians should pay careful attention to expectant women if they show strong stigma towards COVID-19.

We should consider the limitations of this study. We developed a statistically robust measure of COVID-19 stigma. However, this was limited to a population of pregnant women. We should exercise caution in extrapolating the data. Because of the research design, the participants were limited to those in the first trimester. Results may be different in women in the second or third trimester. Issues such as stigma may be influenced by social desirability. Our results may be biased and the participants may underestimate their attitudes.

Taking into consideration these drawbacks, however, the instrument we developed is an easy-to-use, statistically robust measure of stigma against COVID-19 and people infected with the virus. The present study revealed the multifaceted nature of stigma against COVID-19.

## Figures and Tables

**Figure 1 behavsci-12-00257-f001:**
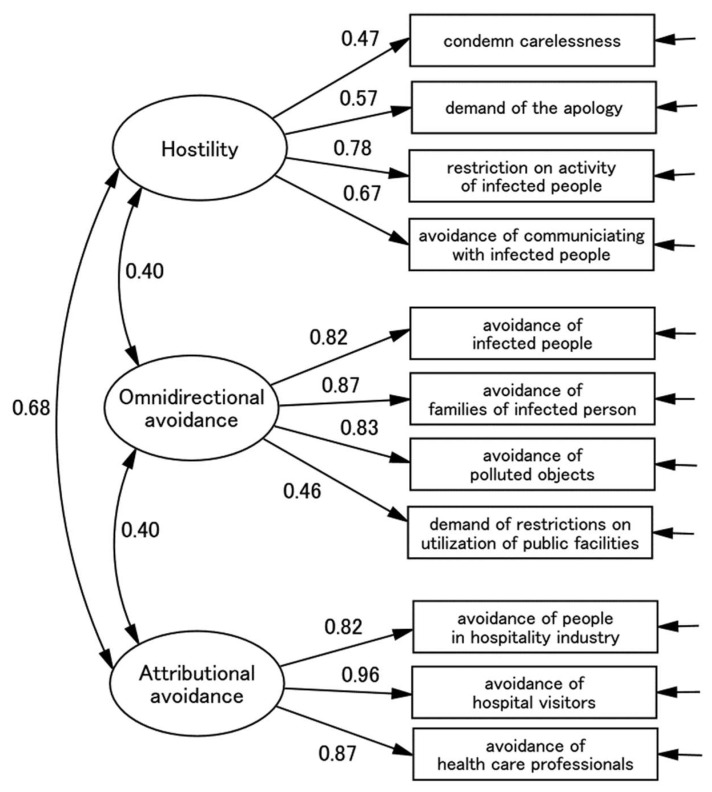
Confirmatory factor analysis of the scale items (*n* = 346). CFI, comparative fit index; RMSEA, root mean square error of approximation; AIC, Akaike information criteria. Paths are standardised. The names of error variables are not shown.

**Table 1 behavsci-12-00257-t001:** Demographic characteristics (*n* = 697).

	Mean	SD
Age	31.7	4.53
Gestational age (weeks)	13.4	1.14

	*n*	%
Gravidity		
0	394	56.6
1 time	169	24.3
2 times	79	11.4
3 times	34	4.9
4 times	12	1.7
5 times	6	0.9
6 times	2	0.3

Parity		
Nuliarae	512	73.6
Multiparae	184	26.4
1 time	126	18.1
2 times	41	5.9
3 times	14	2.0
4 times	3	0.4

Education		
Secondary school	17	2.4
High school	132	19.0
Junior college or Vocational school	192	27.6
Bachelor’s	320	46.0
Master’s	32	4.6
Doctorate	3	0.4

Infertility treatment		
None (spontaneously)	466	67.0
Intercourse timing therapy	149	21.4
Assisted conception	81	11.6
	Mean	SD
Treatment duration (years)	0.29	0.72
	*n*	%
Student		
Yes	6	0.9
No	690	99.1

Employment		
Unemployed	83	11.9
Temporary work	109	15.7
Full-time employment or Self-employed	504	72.4

Have a partner		
Yes	690	99.1
No	6	0.9

**Table 2 behavsci-12-00257-t002:** Mean, SD, skewness, kurtosis of the COVID-19 stigma scale items (*n* = 350).

Item No.	Items (Abbreviations)	Mean (SD)	Skewness	Kurtosis
1	People get infected because they are careless. (condemn carelessness)	2.03 (1.48)	0.06	−0.92
2	Those who are infected/positive should apologise. (demand for apology)	0.56 (1.07)	1.96	3.34
3	I do not want to get close to those who are infected/positive. (avoidance of infected people)	4.97 (1.32)	−1.59	2.92
4	I do not want to get close to the families of those who are infected/positive. (avoidance of families of infected person)	4.34 (1.50)	−0.82	0.47
5	I do not want to touch anything touched by those who are infected/positive. (avoidance of polluted objects)	4.56 (1.53)	−1.12	1.08
6	I do not want to get close to those who are in the hospitality industry. (avoidance of people in the hospitality industry)	1.55 (1.50)	0.65	−0.24
7	I do not want to get close to those who visit hospitals. (avoidance of hospital visitors)	1.40 (1.47)	0.82	−0.18
8	I do not want to get close to health care professionals. (avoidance of health care professionals)	1.14 (1.42)	1.05	0.17
9	Those who were infected/positive should not go to workplace (school) even after they are cured. (restriction on activity of infected people)	1.25 (1.49)	1.01	0.09
10	Those who are infected/positive should not use public transportation or public places. (restrictions on utilisation of public facilities)	4.12 (1.99)	−0.89	−0.34
11	I do not think I will be able to associate as before with those who are infected/positive. (avoidance of communicating with infected people)	0.84 (1.24)	1.49	1.70

**Table 3 behavsci-12-00257-t003:** EFA of the COVID-19 scale items (*n* = 350).

Item No.	Item Contents (Label)	1-Factor	2-Factor		3-Factor			4-Factor			
		I	I	II	I	II	III	I	II	III	IV
1	condemn carelessness	**0.33**	0.22	0.20	0.17	0.03	**0.30**	0.11	0.02	0.03	**0.44**
2	demand for apology	**0.43**	**0.35**	0.13	0.08	0.08	**0.43**	−0.05	0.01	0.03	**0.80**
3	avoidance of infected people	**0.37**	−0.02	**0.83**	**0.85**	−0.00	−0.07	**0.83**	−0.00	−0.11	0.07
4	avoidance of families of infected person	**0.45**	0.06	**0.83**	**0.84**	0.05	−0.02	**0.85**	0.05	−0.01	−0.02
5	avoidance of polluted objects	**0.42**	0.01	**0.86**	**0.86**	−0.05	0.06	**0.87**	−0.04	0.06	−0.01
6	avoidance of people in hospitality industry	**0.80**	**0.80**	−0.00	0.02	**0.75**	0.05	0.00	0.76	−0.03	0.11
7	avoidance of hospital visitors	**0.93**	**0.97**	−0.04	−0.02	**1.01**	−0.03	−0.02	**1.01**	−0.03	−0.01
8	avoidance of health care professionals	**0.88**	**0.89**	−0.02	0.01	**0.83**	0.05	0.02	**0.85**	0.07	−0.08
9	restriction on activity of infected people	**0.55**	**0.50**	0.07	−0.02	0.08	**0.69**	0.02	0.09	**0.67**	0.00
10	restrictions on utilisation of public facilities	0.29	0.11	**0.37**	**0.34**	−0.01	0.18	**0.36**	−0.00	0.21	−0.05
11	avoidance of communicating with infected people	**0.49**	**0.44**	0.07	−0.04	−0.05	**0.81**	−0.00	−0.03	**0.75**	0.05

Factor loadings > 0.3 are in bold.

**Table 4 behavsci-12-00257-t004:** Comparison of EFA-derived factor models (*n* = 347).

Models	*χ*^2^/*df*	D*χ*^2^ (*df*)	CFI	DCFI	RMSEA	DRMSEA	AIC
Models derived from EFA							
1-factor	905.466/46 = 19.684	Ref	0.515	Ref	0.233	Ref	967.466
2-factor	314.468/44 = 7.147	590.998 (2) ***	0.847	0.332	0.133	0.100	380.468
3-factor	111.439/41 = 2.718	203.029 (3) ***	0.960	0.113	0.071	0.062	183.439

*Note.* CFI, comparative fit index; RMSEA, root mean square error of approximation; AIC, Akaike information criteria. *** *p* < 0.001.

**Table 5 behavsci-12-00257-t005:** Configuration, measurement, and structural invariances of the 3-factor model between nulliparae (*n* = 512) and multiparae (*n* = 185).

	*χ* ^2^	*df*	*χ*^2^/*df*	Δ*χ*^2^ (*df)*	CFI	ΔCFI	RMSEA	ΔRMSEA	AIC	Judgement
Configuration	198.317	82	2.419	Ref	0.968	Ref	0.045	Ref	342.317	ACCEPT
Metric	205.422	90	2.282	7.015(8)NS	0.968	0.000	0.043	0.002	333.422	ACCEPT
Scalar	219.989	101	2.178	14.567(11)NS	0.967	0.001	0.041	0.001	325.989	ACCAPT
Residual	244.051	112	2.179	24.062(11) *	0.964	0.003	0.041	0.000	328.051	ACCEPT
Factor variance	248.448	115	2.160	4.397(3) NS	0.963	0.001	0.041	0.000	326.448	ACCEPT
Factor covariance	253.443	228	2.148	5.365(113) NS	0.963	0.001	0.041	0.000	325.813	ACCEPT

* *p* < 0.05; NS, not significant; CFI, comparative fit index; RMSEA, root mean square of error approximation; AIC, Akaike information criterion.

**Table 6 behavsci-12-00257-t006:** Configuration, measurement, and structural invariances of the 3-factor model between age < 32 (*n* = 344) and age ≥ 32 (*n* = 353).

	*χ* ^2^	*df*	*χ*^2^/*df*	Δ*χ*^2^ (*df)*	CFI	ΔCFI	RMSEA	ΔRMSEA	AIC	Judgement
Configuration	213.294	82	2.601	Ref	0.964	Ref	0.048	Ref	357.294	ACCEPT
Metric	220.685	90	2.452	7.391(8)NS	0.964	0.000	0.046	−0.004	348.685	ACCEPT
Scalar	234.256	101	2.319	13.571(11)NS	0.963	0.001	0.044	−0.002	340.256	ACCAPT
Residual	253.925	112	2.267	19.669(11) *	0.961	0.002	0.043	−0.001	337.925	ACCEPT
Factor variance	259.720	115	2.258	5.795(3)NS	0.960	0.001	0.043	0.000	337.720	ACCEPT
Factor covariance	267.575	118	2.268	7.855(3) *	0.959	0.001	0.043	0.000	339.575	ACCEPT

* *p* < 0.05; NS, not significant; CFI, comparative fit index; RMSEA, root mean square of error approximation; AIC, Akaike information criterion.

**Table 7 behavsci-12-00257-t007:** Factor mean of the 3-factor model.

	Factor Mean (SE)		
	F1: Omnidirectional Avoidance	F2: Attributional Avoidance	F3: Hostility
Nulliparae (*n* = 512) compared with multiparae (*n* =184)	−0.070 (0.082) NS	−0.199 (0.115) NS	0.034 (0.359) NS
age less than 32 years (*n* = 344) compared with age 32 years or older (*n* = 353)	−0.122 (0.074) NS	−0.049 (0.101) NS	−0.052 (0.083) NS

NS, not significant; SE, standard error.

**Table 8 behavsci-12-00257-t008:** Correlations of Omnidirectional avoidance, Attributional avoidance, and Hostility with predictor variables.

	Omnidirectional Avoidance	Attributional Avoidance	Hostility
Demographic and obstetric variables			
Age	0.06	0.04	0.02
Gestational age	0.1	0.2	0.7
Past pregnancy (times)	−0.04	0.00	−0.06
Past childbirth (times)	−0.05	0.04	−0.04
Preventive means against COVID-19			
Mask	−0.02	−0.06	−0.13 **
Hand washing	−0.03	−0.07	−0.12 **
Gargling	−0.06	0.01	−0.03
Showering	−0.07	−0.02	−0.05
Alcohol disinfection	−0.03	−0.04	−0.05
Gloves	−0.13 **	−0.04	−0.08 *
Faceguard	−0.10 **	−0.04	−0.01
Attitude towards the present pregnancy			
Denial of pregnancy	−0.03	0.01	0.04
Unintended pregnancy	−0.06	−0.03	0.03
Quit caring	−0.04	0.10 **	0.04
Wish to terminate	0.01	0.07	0.10 **
Current pregnancy			
PUQE Total	0.00	−0.04	0.06
Perceived impact of pregnancy	0.05	−0.00	−0.10
Assisted conception (yes, 1; no, 0)	0.09	−0.00	0.05
Infertility treatment duration (years)	0.07	0.03	00.05
Mental state and psychopathology			
Fear of child birth	0.04	0.13 ***	0.15 ***
Foetal bonding			
Happiness	0.03	−0.09 *	−0.15 **
Anger	−0.04	0.09 *	0.13 **
Fear	0.01	0.07	0.15 ***
Sadness	−0.01	0.12 **	0.18 ***
Disgust	−0.03	0.10 **	0.16 ***
Surprise	0.01	0.04	0.17 ***
Shame	−0.03	0.10 **	0.14 ***
Guilt	−0.10	0.04	0.12 **
Alpha pride	0.02	0.02	0.04
Beta pride	0.05	−0.01	−0.07
Substance use			
Smoking amount	0.00	0.01	0.02
Alcohol	−0.05	−0.06	−0.07
Obsessive compulsive symptoms			
Washing	0.17 ***	0.31 ***	0.31 ***
Checking	0.03	0.23 ***	0.21 ***
Ordering	0.12 **	0.21 ***	0.24 ***
Obsession	0.11 **	0.19 ***	0.24 ***
Hoarding	0.07	0.15 ***	0.23 ***
Neutralising	0.06	0.20 ***	0.28 ***
MDE			
Depression	0.08 *	0.13 **	0.15 ***
Anhedonia	0.04	0.14 ***	0.15 **
Total	0.06	0.14 ***	0.15 ***
Adult attachment			
Self-model	0.02	−0.14 ***	−0.15 ***
Other-model	0.08 *	−0.08 *	−0.05
Borderline personality traits			
Primitive difences	0.03	0.11 **	0.24 ***
Identity delusion	0.06	0.11 **	0.21 ***
Reality testing	0.01	0.20 ***	0.21 ***
Total	0.04	0.16 ***	0.26 ***

* *p* < 0.05; ** *p* < 0.01; *** *p* < 0.001.

## Data Availability

The datasets used and analysed in the present study are available from the corresponding author upon reasonable request.

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
