# Peer review of "Development of a Scale for COVID-19 Stigma and Its Psychometric Properties: A Study among Pregnant Japanese Women"

_behavsci, 2022, doi:10.3390/bs12080257_

Round 1
Reviewer 1 Report
A very important topic to measure. My only observation is that need to report multivariate kurtosis, the program, and estimator used in CFA.
Author Response
Table 2 presents skewness and kurtosis of all the items of the scale. The reviewer pointed out multivariate kurtosis. We appreciate that this is a very important point. However, we cannot examine multivariate kurtosis with our current command of SPSS or AMOS. In a future paper, instead, we plan to examine this issue from a toxometric perspective. This will be another paper. We added details of our procedures related to CFA.
Reviewer 2 Report
The study examined the Development of a Scale for COVID-19 Stigma and its Psycho-2 metric Properties: A Study among Pregnant Japanese Women. The topic sounds interesting. There is some point need further explanation, for example; why choose the pregnant women only? It is confused about the rationales for this study. I would recommend minor revisions to the author, there are some obvious issues that need revision. At the same time, are there any scales or studies for this stigma before Covid-19? Overall, it is very interesting topic and would recommend minor revisions
Author Response
We added the following sentences in Introduction.
When searching PubMed with ((stigma) AND (COVID-19)) AND (Scale OR inventory OR Measurement)), a total of 339 papers were hit. Of these, we identified 14 papers delt with COVID-19 stigma. However, none of them treated the issue in psychometric details (e.g., content validity, factor structure, and measurement invariance).
Reviewer 3 Report
I would like to congratulate the research team for such a novel and timely approach to the COVID situation. This indeed seems to be the first COVID-specific stigma questionnaire in pregnant women only. First stigma forming needs to be identified before something can be done about it. This is especially important in socially challenging times such as the COVID pandemic. I have no further recommendations to make.
Author Response
Thank you for your comments.